# An Investigation on the Effect of Leakages on the Water Quality Parameters in Distribution Networks

Daniel Barros [1,*,†], Isabela Almeida [1,†], Ariele Zanfei [2,†], Gustavo Meirelles [1], Edevar Luvizotto, Jr. [3] and Bruno Brentan [1,†]

1 Hydraulic Engineering and Water Resource Department, Federal University of Minas Gerais, Belo Horizonte 31270, Brazil
2 Faculty of Science and Technology, Free University of Bozen-Bolzano, Piazza Università 5, 39100 Bolzano, Italy
3 Laboratory of Computational Hydraulics, University of Campinas, Campinas, São Paulo 13083, Brazil
* Correspondence: danielbezerrab@gmail.com
† These authors contributed equally to this work.

**Abstract:** Leakages in distribution networks reach more than 30% of the water supplied, entailing important risks for the water infrastructure with water contamination issues. Therefore, it is necessary to develop new methods to mitigate the amount of water wastes. This study proposes to seek new sources of information that can help for a more sustainable water use. Hence, an analysis of the network is presented, showing the hydraulic behavior during leaks occurrence, placing emphasis on how these events affect and modify water quality parameters, such as water age and chlorine concentration. The study enhances that water quality data can be an effective source of information in the case of leaks, being a possible source of information for future detection systems. In addition, this study proposes to use graph theory on the water network. The results highlight how an analysis of the shortest path between the leak location and the reservoir could provide meaningful information for future detection systems.

**Keywords:** water distribution network; leak detection; water quality; anomaly detection; water supply



## 1. Introduction

Water is a fundamental resource for life, which is why, in 2000, at the United Nations General Assembly, 191 countries agreed on seven main Millennium Development Goals (MDG) [1]. Among them, the fundamental target of MDG-7 was to halve the proportion of people without sustainable access to safe drinking water and basic sanitation by 2015 [2]. According to the website MDG Brazil (2021), this goal was achieved in advance by the country, but the lack of water, sanitation, and hygiene is closely linked with other MDGs, such as MDG-4, which aims to reduce child mortality. According to Howard et al. [3] the lack of water, sanitation, and hygiene is the cause of several diseases that are behind the deaths of children around the world. Death from diarrhea attributed to insufficient water supply and sanitation is ranked as the sixth most prevalentdisease that most kills children. At this point, Brazil still registers an infant mortality rate (under one year old) of 15.6 and a child mortality rate (under 5 years old) of 19 deaths per thousand births [4].

Although in 2000 1.1 billion people in the world did not have access to quality water and its countless benefits, paradoxically, the waste and losses of treated water in water distribution networks (WDNs) in several countries are a real problem. According to the World Health Organization (WHO), large cities in Africa, Asia, Latin America, the Caribbean, and North America have an average loss of water in the distribution system of 35%, with Latin America leading this front with 42% [5]. Rebouças [6] estimates that this loss may be even greater in Brazil, reaching 60% in some cities, against the benchmark values of 5% to 15% in developed countries.

In the report produced by Trata Brasil [7] on water losses caused by leaks in pipes, lack of flow meters, measurement errors, clandestine connections, and water theft in Brazil, in 2018, the losses represented 38.45% of water volume in the distribution. The study was obtained from the database of the National Sanitation Information System (SNIS) and showed that losses are equivalent to 6.5 billion cubic meters. This value represents an average loss in total revenue of 39.02%, about BRL 12 billion for service providers (water and sewage). It is worth noting the great social impact of these losses, whose wasted volume could supply the 13.5 million people who currently reside in favelas for approximately 2 years.

In particular, it is worth mentioning the measured physical losses, which are losses caused by leaks in the network and branches, structural leaks, overflows, and discharges during the processes of raw water adduction, treatment, storage, treated water adduction, and distribution in the network, which directly affect the relation between demand and production of water, which resulted in values close to 3.4 billion cubic meters of water lost.

Physical losses represent an increase in the energy cost of pumping water, overuse of production and distribution systems, and higher cost for managing the environmental impact of the activity [7]. Detecting these leaks is essential to ensure reduction of these impacts. For this, different methods are used; one of these, for example, is ground penetration radar. This method consists of analyzing cross-sectional soil profiles around the pipes to detect water leaks [8].

The ground penetration radar method is named as an external detection approach, as well as the acoustic methods that aim to identify leaks by anomalies in sound waves that are able to travel through pipes and/or surrounding surface when pressurized water leaks through an orifice [9,10]. These techniques are capable of identifying even small leaks, but they have several disadvantages: high time consumption, difficult application in large areas, dispersion of the acoustic signal, etc. [8,11].

The second major category of leak detection uses techniques that are continuously monitoring, through sensors, internal network parameters such as water velocity or pressure [8,12,13]. Taking advantage of the large number of measurements that can be provided, the inverse transient analysis (ITA) method uses data to simulated transient events in pipes, looking to improve leak detection [14,15]. A variation of the ITA is the pressure flow bypass method. A leak is declared if there are pressure deviations between the two edges of the pipe. The efficiency of these models, as well as of several other internal detection techniques, depends on the efficient location of the sensors and mainly on the level detail of models to reduce the errors between simulations and the actual measured values [12].

Continuous monitoring generates a large amount of data and this allows the application of techniques related to data mining. These techniques are called data-driven approaches and they look for outliers that deviate from data behavior patterns and associate them with anomalies. Wu and Liu [16] classify them into three categories: classification methods, predictive classification methods, and statistical methods.

Classification methods generally use data-driven approaches to learn the normal behavior of the network, and thus are able to recognize and classify abnormal events such as leaks [17]. Predictive classification methods differ from the classification, and rely on the availability of normal hydraulic data (i.e., not affected by anomalies). The normal data are therefore used to create a model for making predictions, and use techniques such as Kalman filter [18] and support vector machine [19] to predict the data value, and if they have a difference greater than a given threshold, an anomaly is detected. Finally, statistical methods directly use the discrepancies caused by leakage in the measured data to detect anomalies. For this, methods such as standard deviations [20] and independent component analysis [21] are used in the literature.

The monitoring methods presented so far for leak detection purposes only consider hydraulic data, such as pressures and flow rates. However, there is also a growing concern about the quality of distributed water, such as water contamination by organic and nonorganic pollutants such as arsenic, copper, pesticides, and trihalomethanes [2]. Bangladesh

had the biggest fight against contaminated water in history. The country was plagued with the contamination of its surface waters by microorganisms, which caused acute gastrointestinal disease, a range of other diseases, and even death, affecting mainly children. In 1970, the United Nations Children's Fund (UNICEF), together with the Department of Public Health Engineering, mobilized to build tube wells. In 1983, the first arsenic contamination patients were identified, and several studies to identify the magnitude of the situation were carried out in universities and laboratories. It is estimated that about 42 million people have been exposed to concentrations above 10 µg/L, which is the maximum level recommended by the WHO. Symptoms of arsenic contamination include skin lesions and different types of cancer (bladder, lung, liver, and kidneys) [22].

Although the mass contamination in Bangladesh was not caused by a direct intrusion into a WDN, the risk cannot be underestimated. Fox et al. [23] demonstrated that an external contamination can enter the network and remain at the point of generating quality drops during short-term transient pressure events. In [23,24], it was shown that the presence of contaminants below the network should also not be neglected, as the influence zone does not depend on its position. The authors emphasized a greater dependence on the distance between the contaminant and the entry orifice, the porous medium around the pipe, and the orifice's size.

The contaminants detection in a network is based on the premise that a contaminant injected into a WDN, whether deliberately, accidentally, or naturally, will affect at least one of the monitored parameters [25]. In their work, Perelman et al. [26] demonstrated, using as parameters the total chlorine, electrical conductivity, pH, temperature, total carbon, and turbidity, that even when a parameter does not detect the presence of contamination, other parameters detect it, and therefore the model obtained a satisfactory result. However, to avoid false positives, they assumed that at least two parameters should detect the contamination. Because of this, efficient placement of sensors that monitor these parameters can be a key factor for probability and detection time to ensure the lowest number of affected consumers. Therefore, sensor placement is usually addressed by multiobjective formulation [25].

Given the risks of contaminant intrusion and its massive health consequences, quality must be a factor when analyzing leaks in WDN. According to Kumar et al. [27], due to the chlorination carried out during the adduction and throughout the system, it is possible to correlate the loss of water caused by leaks with the loss of chlorine injected into the network. Therefore, obtaining data from monitoring stations can serve as a basis for identifying any anomaly in chlorine levels. Additionally, due to the dynamic hydraulic flow observed in a network, leaks at different points produce different effects on water quality, reinforcing the potential of this parameter in identifying leaks.

Even if many studies are proposed in the literature for detecting leaks and other physical anomalies in water systems, most of them are focused on processing hydraulic data (e.g., flow, pressure, tank level) instead of using water quality data (e.g., free chlorine concentration, pH, turbidity). Since the water quality parameters are not often as monitored as the hydraulic ones, and the modeling of mass transportation is much more complex than hydraulic equations, water quality parameters are not explored to their full potential. For better understanding the water quality changes due to the presence of leaks in water distribution systems, this work investigates the effects of leaks on water quality parameters and the possibility of using water quality monitoring data for detecting leaks. Thus, this work computationally simulates leaks and conceptually proves that quality data can be an additional source of information for monitoring not only water quality, but also in cases of leaks. For this, Epanet 2.2 and the Python package Water Network Tool for Resilience (WNTR) are used in the simulation processes. Finally, the article also presents a methodology to determine the shortest path traveled by water between the reservoirs and the leak site. This methodology uses concepts linked to graph theory and analyzes changes in flows in pipes due to leaks.

## 2. Materials and Methods

To carry out the water quality study as a leak indicator, EPANET software and the WNTR package [28] in the Python environment are used. EPANET is widely used as a support tool for the analysis of water distribution systems, allowing the execution of steady and extended period simulations of the hydraulic behavior and water quality of pressurized distribution systems [29]. WNTR, in turn, is based on the EPANET program, but its application programming interface is more flexible, allowing for changes in the structure and operations of the network [30].

Knowing that water quality changes due to water loss in cases of leaks [27], this research investigates which are the impacts on quality parameters when simulated leaks occur. Thus, to perform the simulations, two orifice equations are used to modeling the leaks. Both equations are used in leak simulations to determine which one best represents a real leak and its uses in the different stages of this research. The simulations are performed varying parameters of the leak, such as its orifice size, the duration of the leak, and form of start. These parameter variations allow to have more simulation scenarios and to observe the influence of different leakages on the water quality changes.

### 2.1. Leakage Mathematical Modeling

The simulations are performed using the standard equation of the EPANET 2.2 software. For the leak simulation in WDN with EPANET, emitter devices are used. According to [29], such devices are associated with junctions that model the flow through orifices or nozzles with direct discharge to the atmosphere. The flow through these devices varies depending on the pressure at the junction, according to a flow law of the type:

$$q = C_e \cdot P^y \tag{1}$$

where $q$ is the flow rate, $P$ is the pressure head, $C_e$ is the discharge coefficient, and $y$ is the pressure exponent.

After the standard simulations, one other equation is used in order to compare their influence on leak detection through quality; the equation is the standard orifice equation [24]:

$$q = C_e \cdot A\sqrt{2gP} \tag{2}$$

where $A$ is the orifice area and $g$ is the gravity acceleration.

The orifice equation is chosen since the simulation in EPANET mode using the WNTR package does not allow the inclusion of leaks in specific periods of the simulation. Therefore, the orifice equation is used to determine the leak flow at specific times of the simulation, using the hourly pressure and including it with additional demand on the network nodes.

### 2.2. Simulation Process

Leakage simulations are made, proposing three different scenarios where the location and duration of the simulated leaks change. In order to obtain a more realistic, and to perform a more robust, investigation, all leakage scenarios are taken into account at four different nodes during the simulation period. Scenarios A and B present overlapping leaks with a duration of 7 days, while leaks in scenario C do not overlap and have a duration of 14 days. For all simulations, the same orifice diameters are used: 10 mm, 8 mm, 15 mm, and 9 mm, respectively, for the four nodes selected in each scenario. These nodes are named Junction 1 to Junction 04 in each scenario, and parameters assigned to each node are shown in Table 1.

**Table 1.** Leakage scenarios and parameters.

| Scenario A | Junction 01 | Junction 02 | Junction 03 | Junction 04 |
|---|---|---|---|---|
| Nodes | 188 | 122 | 50 | 45 |
| | 110 | 263 | 241 | 49 |
| Start leakage (days) | 1 | 2 | 2 | 5 |
| Duration leakage (hour) | 120 | 144 | 72 | 30 |
| Start form | Three days of raise | Five days of raise | Abrupt | Two days of raise |
| **Scenario B** | **Junction 01** | **Junction 02** | **Junction 03** | **Junction 04** |
| Nodes | 255 | 137 | 29 | 136 |
| Start leakage (days) | 3 | 2 | 1 | 4 |
| Duration leakage (hour) | 24 | 3 | 72 | No end |
| Start form | One day of raise | Abrupt | Two days of raise | Seven days of raise |
| **Scenario C** | **Junction 01** | **Junction 02** | **Junction 03** | **Junction 04** |
| Nodes | 213 | 45 | 150 | 156 |
| Start leakage (days) | 2 | 5 | 7 | 12 |
| Duration leakage (hour) | 60 | 21 | 96 | 67.2 |
| Start form | Two days of raise | Abrupt | Three days of raise | One days of raise |

All parameters in Table 1 are chosen randomly and will be used as they appear, only changing the nodes with leaks. Scenarios can also be repeated, but with the change of nodes where leaks occur. These variations in sites and parameters allow variability and a certain range of simulated leaks and improve the investigation process.

*2.3. Nodes' Sensitivity to Leakage*

A sensitivity analysis is performed to determine which nodes or regions in the network are most sensitive to leaks. This will allow the determination of possible monitoring points, which will be used to expose the results in this research. For this, initially, two simulation processes are performed. The first lasts 24 h, considering the network without leakages and saving hourly pressure and quality data. The second process adds a leak to a specific junction and studies its influence on the other network junctions. This process is repeated until all junctions are simulated with a leak. Each simulation lasts 24 h and the pressure and quality data from each junction are used to create the hourly sensitivity matrices ($S_{ij}$) determined by the following equation:

$$S_{ij} = \frac{D_i - D_i^*}{q_j} \tag{3}$$

where $D_i$ is the pressure or quality data of the node *i* without leaks, $D_i^*$ is the pressure or quality data with leakage, and $q_j$ is the leak flow rate in node *j*. This process results in two square matrices, one for pressure and another for quality. Through the sensitivity matrices, it is possible to determine the nodes with pressure and quality most influenced by leaks. Thus, the analysis of data from the most sensitive nodes enables a greater understanding of the behavior of pressure and quality.

After checking the sensitivity of each node, a new simulation process is performed with a different leak, following the parameters depicted in Table 1.

In the present paper, the water age is used initially as a quality parameter to be observed. The age of water can be determined by the EPANET software and has supported computational modeling research in water networks [31]. The pressure and quality of the most sensitive nodes are monitored in order to identify the behavior of these parameters under different leak scenarios. Thus, using mainly the daily average of pressure and quality, it is possible to mathematically identify its alteration.

*2.4. Graph Theory and Shortest Path*

To study in more detail the relationships between leakage events and quality variations, it is possible to use the graph theory, which is a mathematical approach that identifies the interactions between objects. A graph is represented by $G = (V, E, W)$, where $V$ portrays the vertices of the graph, and these vertices are the representation of the objects; $E$ are the edges of the graph, representing the connections between vertices; and $W$ are the edge weights, characterizing stronger or weaker connections between vertices. A WDS can be represented by a graph considering the nodes as the vertices and the edges as the pipes. The weights of the edges can be the flow rates, head losses, or roughness of the pipes [32,33].

In the present work, the graph theory is used to determine the shortest paths between the nodes and the reservoir, considering the maximum flow rates of the pipes as edge weight. Hence, a graph structure is created, starting from the network topology, where the graph vertices represent the network nodes, while the graph edges represent the pipes. Furthermore, at each edge, a weight is assigned that is the maximum flow rate of the related pipe. This approach uses the Network Analysis in Python (NetworkX) Python package [34], which allows the creation, manipulation, and study of dynamic structures of complex networks and graphs.

The shortest path determination in a graph starts by evaluating a starting point and a destination point. After this calculation, the distances between the edges to the neighboring vertices of the starting point are calculated. This distance can be considered using the smallest number of edges to the destination vertex, the edge weights, or other factors attached to the edges. The method determines which neighbor vertex has the shortest distance to the starting point, and then performs the same process for the determined neighbor vertex, always directing it to the destination vertex. At the end of the process, the edges and vertices belonging to the shortest path are obtained [35].

*2.5. Evaluation Method*

In order to quantify the effect of different leaks on water quality parameters, it is proposed to use two evaluation methods: firstly, the use of maps to associate variation in the analyzed parameters with the spatial characteristics of the network; secondly, to calculate variation in both hydraulic parameters and quality parameters from the scenarios affected by leaks, and the normal behavior of the network (i.e., without leaks). Particularly, to emphasize the variations in the different parameters analyzed, it is proposed to calculate a *Delta* parameter defined as the percentage difference between the value in the scenario affected by leaks and the normal condition. This parameter is defined as:

$$Delta = \frac{D - D^*}{D} * 100 \qquad (4)$$

where $D$ is the pressure or quality data without leaks, and $D^*$ is the pressure or quality data in the scenario with leakages. Thus, it is possible to highlight and quantify the modification in the various parameters with respect to the presence of a leak or not.

**3. Case Study**

The network used for this study (Figure 1) is presented in the work of Bragalli et al. [36], based on the city's distribution network in the Emilia-Romagna region of Italy, Modena. It has a total of 272 nodes, of which 268 are junctions and 4 are reservoirs. In addition, it features 317 pipes and has no valves or pumps.

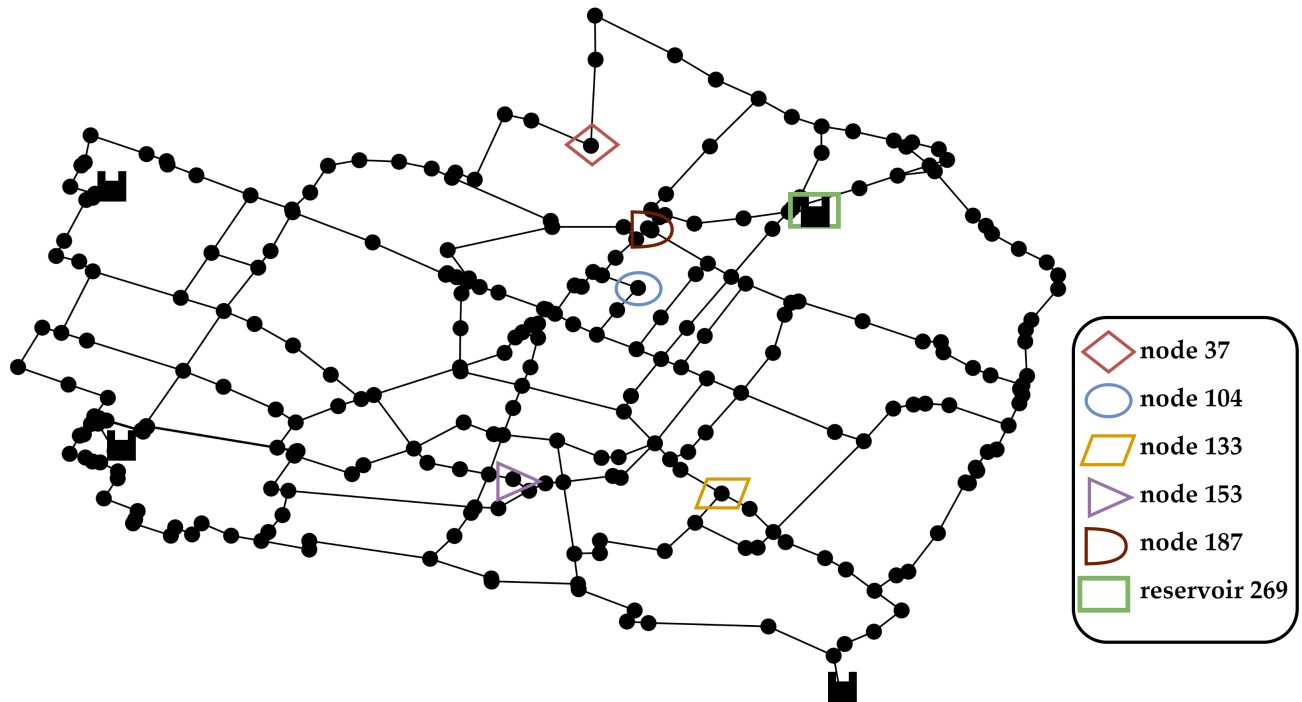

**Figure 1.** Modena network.

Figure 1 presents the Modena network and highlights some nodes and a reservoir, which will be explored in the course of this paper. This network has been widely used in a plethora of problems, such as sensor placement [37], leak detection [33], and energy efficiency [38].

Table 2 reports the maximum flow losses caused by leaks in the different scenarios of the proposed investigation.

**Table 2.** Leakage nodes and flow rates.

| Scenario A | | | |
|---|---|---|---|
| Nodes | 188 | 122 | 50 | 35 |
| Flows (LPS) | 2.55 | 1.45 | 4.34 | 1.31 |
| Nodes | 110 | 263 | 241 | 49 |
| Flows (LPS) | 2.36 | 1.38 | 4.47 | 0.98 |
| **Scenario B** | | | |
| Nodes | 255 | 137 | 29 | 136 |
| Flows (LPS) | 2.00 | 1.59 | 4.78 | 0.83 |
| **Scenario C** | | | |
| Nodes | 213 | 45 | 150 | 156 |
| Flows (LPS) | 2.23 | 1.52 | 3.39 | 1.89 |

The maximum leakage flows range from 0.89 to 4.78 LPS, which represent small leaks and grids according to Quiñones-Grueiro et al. [39]; this represents the inclusion of different amount of leakages in terms of magnitude. Comparing these values to total consumption of the network (406.94 LPS), this would mean leakage at 0.2% to 1.17% of total consumption.

## 4. Results

### 4.1. Sensitivity Analysis

To evaluate the simulation results graphically, as mentioned above, two average sensitivity matrices are created, one for the water quality and another for the pressure. These matrices are constructed with the rows representing the nodes with simulated leak and the columns representing the sensitivity data of the other network nodes. This is made possible through Equation 3. Then, the average value for each column of the matrix is calculated, resulting in an average value for each node for both quality and pressure. These values are shown through the map in Figure 2.

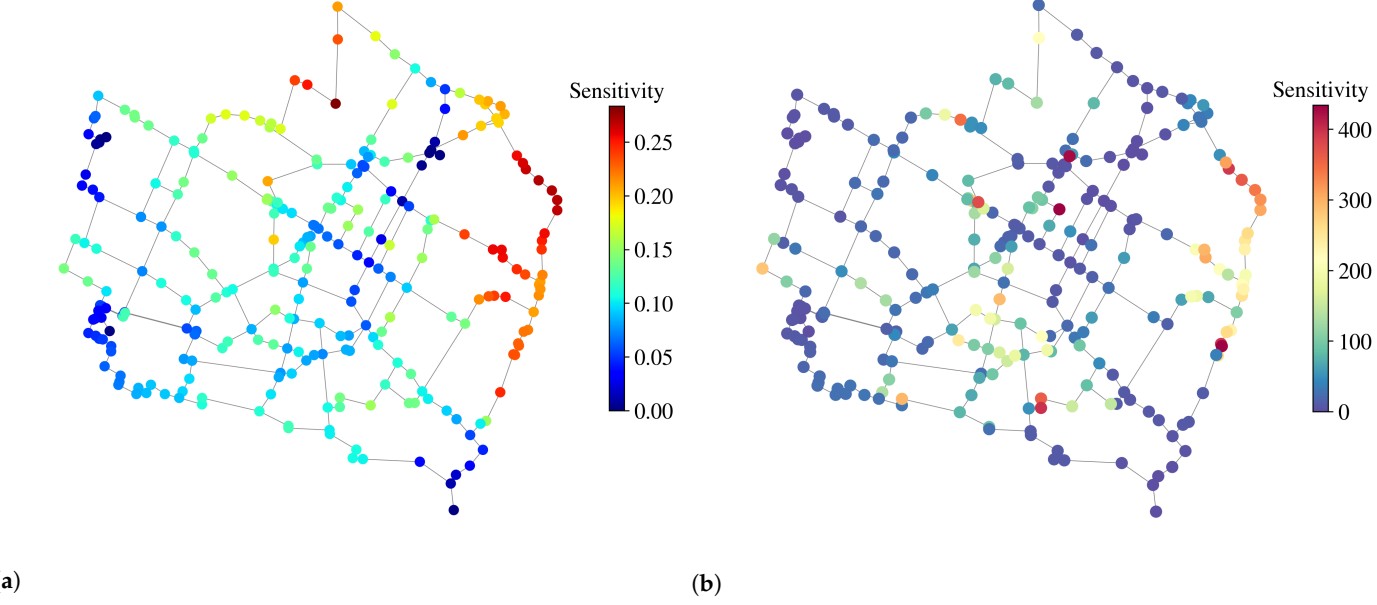

(**a**)                (**b**)

**Figure 2.** Sensitivity map: (**a**) pressure and (**b**) quality.

The sensitivity for both parameters (Figure 2) is lower at nodes around the reservoirs. This happens because the reservoirs are at a fixed level (constant hydraulic grade) since they supply water directly to these nodes (low headloss influence). Furthermore, the map allows to highlight some nodes that exhibit particularly different sensitivity values. A notable point is junction 104, which is highlighted in red in Figure 2b. This node exhibits high sensitivity for all scenarios, with or without overlapping leaks during the simulation period.

It is proposed to assign junctions 188, 122, 50, and 45 to the parameters of junctions 01, 02, 03, and 04 of scenario A to make an additional test. Figure 3 shows the variation of pressure and quality in node 104.

For the simulated leak scenario, the nodes had leakage flows following Table 2. Even with the highest tested leak flows (4.34 *LPS*), the pressures in the sensor nodes exhibit a low variability. If it is considered to be much higher flows than those tested, the entire network behavior is modified and can be easily identified by pressure and quality data. In contrast, the water age in the sensor node was significantly affected by small or big leakages. It is observed that the Delta parameter for the pressure data (Figure 3b) changes slightly more than 1%, while the Delta for the quality data have changes greater than 500% (Figure 3d). This shows that although the pressure is effective on the network coverage to detect leaks, quality data can help to increase this value or reduce the sensors number. The study highlights that the use of the quality sensor can provide important support in leak detection, given the high variability shown during leak events.

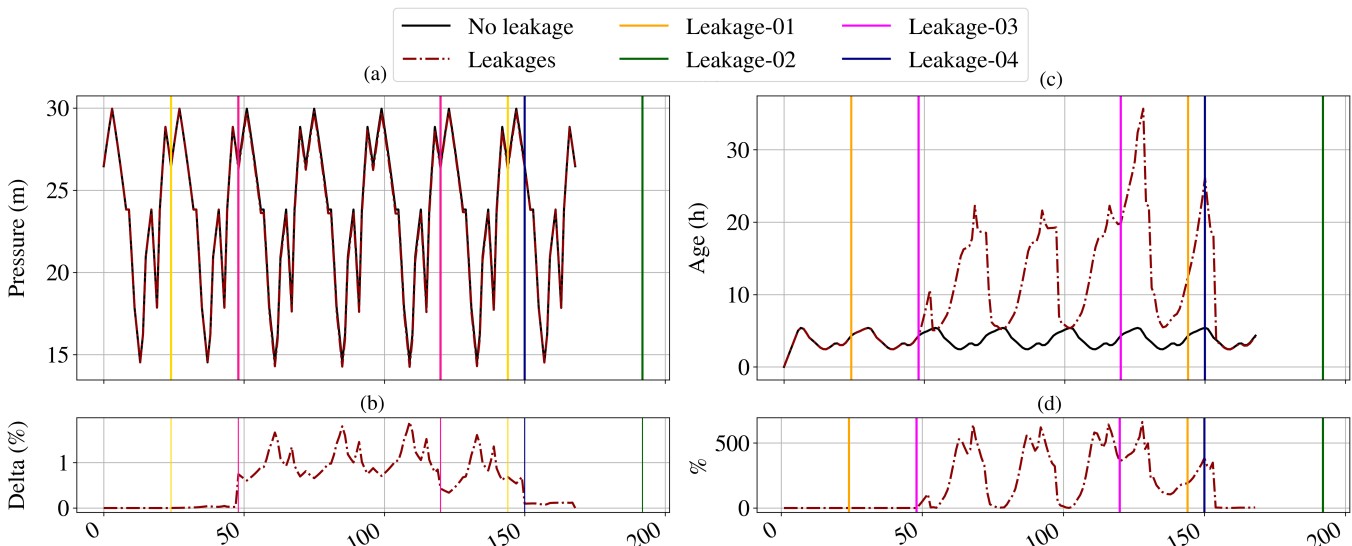

**Figure 3.** Node 104 - Monitoring data: (**a**) Quality variation, (**b**) Quality variation delta, (**c**) Pressure variation and (**d**) Pressure variation delta.

### 4.2. Pressure and Quality

After the general analysis of the entire network, shown in Figure 2, simulations are carried out considering the leakage in specific nodes following Table 1. To analyze the influence of simulated leaks in the network, some nodes are also chosen to monitor their quality and pressure values. In particular, nodes 104 and 187 stand out due to their higher-quality sensitivity and, therefore, are used as sensor nodes in the next stages of this research.

Considering the parameters of scenario A, and assigning them the properties of "Junction 01", "Junction 02", "Junction 03", and "Junction 04", to nodes 110, 263, 241, and 49, respectively, Figure 4 highlights the variations of quality and pressure for the monitoring node 104.

It is worth noting that in Figure 4 the pressure behavior also does not exhibit significant variations. On the other hand, quality varies consistently, showing fluctuations of almost 40% of its value (Figure 4d), reinforcing its possible role in helping to detect leaks. Other scenarios reinforce the role of quality as a future ally for detecting leaks in networks. As further confirmation, it is shown in Figure 5 the results related to scenario B, linking the parameters of junctions 01, 02, 03, and 04 to nodes 255, 137, 29, and 136, respectively. In particular, Figure 5 shows the variations of the monitoring node 187.

Finally, Figure 6 proposes the results for scenario C that assigned to nodes 213, 45, 150, and 156 the parameters of junctions 01, 02, 03, and 04 presented in Table 1. Node 153 is used as monitoring node.

Figure 2b highlights that the water age changes due to leaks because, to meet the demand and leakage, the water travels different paths, in some cases increasing the water velocities in the pipes or receiving water from different reservoirs. However, water age is not a real monitoring parameter, so a simulation process observing chlorine is proposed afterward.

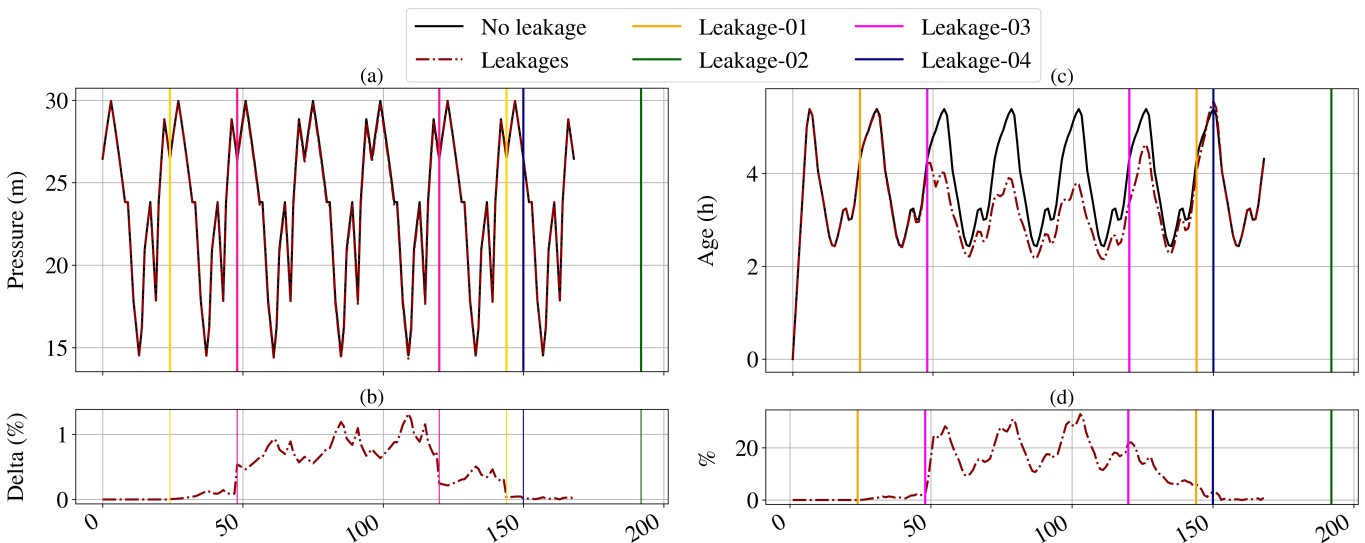

**Figure 4.** Sensitivity of node 104 for leaks in scenario A: (**a**) Pressure variation, (**b**) Pressure variation delta, (**c**) Quality variation and (**d**) Quality variation delta.

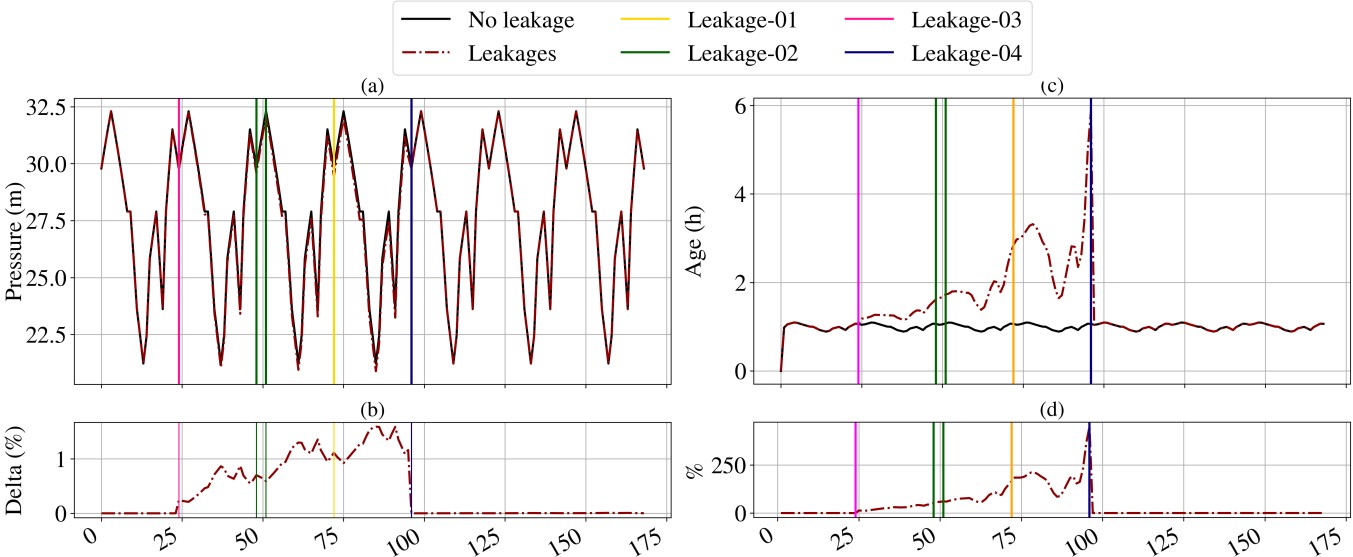

**Figure 5.** Sensitivity of node 187 for leaks in scenario B: (**a**) Pressure variation, (**b**) Pressure variation delta, (**c**) Quality variation and (**d**) Quality variation delta.

### 4.2.1. Chlorine Simulations

Although the water age is a known parameter due to its sensible variations, it does not have a real monitoring method. On the other hand, chlorine might be well monitored and it is used for multiple scopes, for instance, to support the location of contamination points, to determine the water age, and to ensure that the potability standards in terms of concentrations are respected. In addition, Gobet et al. [40] presented a wireless monitoring sensor that has an accuracy of 0.02 mg/L, which promotes leakage detection with a slight change in the concentration of the monitored parameter [41,42].

Hence, in order to identify the variations of the chlorine concentration during leak events, it is proposed to perform simulations considering the reservoirs as sources of chlorine with a continuous supply of water at a concentration of 3.0 mg/L of chlorine. The simulations had the flow reaction coefficient equal to $-2.5$, following the default value for chlorine simulations [29], and the pipe wall reaction coefficient equal to 0.15 [43].

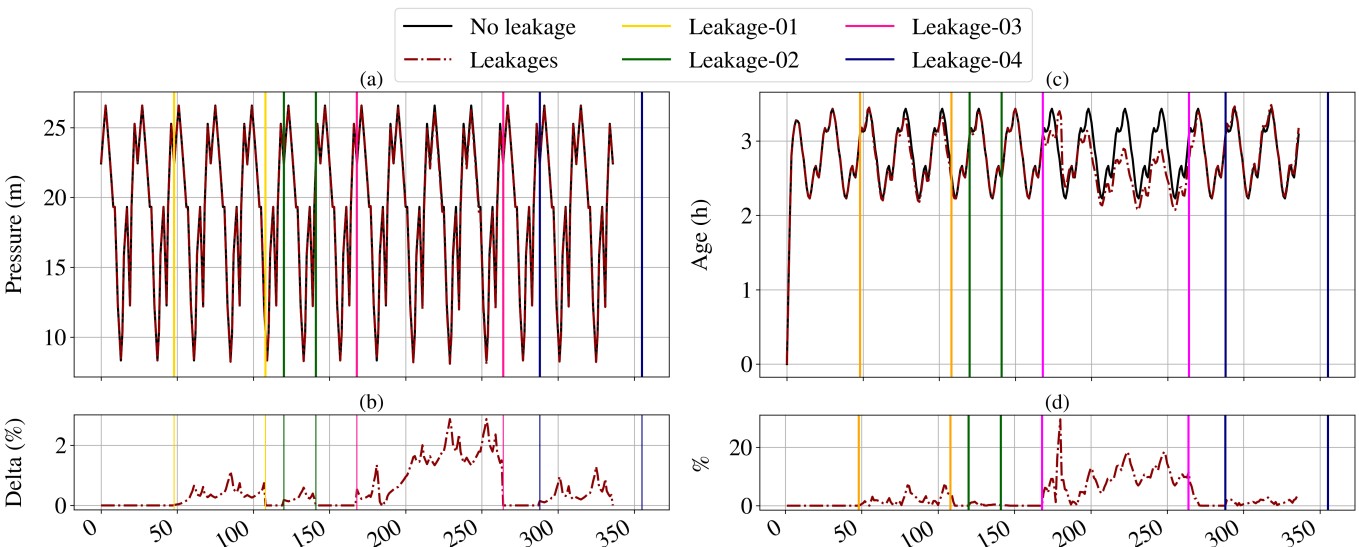

**Figure 6.** Sensitivity of node 153 for leaks in scenario C: (**a**) Pressure variation, (**b**) Pressure variation delta, (**c**) Quality variation and (**d**) Quality variation delta.

As node 104 proved to be dominant for water age sensitivity in the previous scenarios, it is proposed to analyze this node once more. This particular node has a chlorine concentration that is very sensitive to leakages at almost all nodes of the network. Figure 7 highlights the maximum absolute differences of the chlorine concentration values at each node with the node 104. These maximum differences are evaluated considering the simulations with a leak in each node and for their whole duration. For this, a new process of simulations was performed; in this case, a 24 h simulation was performed individually at each node and the maximum chlorine concentration was observed at node 104.

The numerical chlorine variations shown in Figure 7 demonstrate that most of the network nodes' leakage scenarios (86.6%) affect approximately 1.0 mg/L of the chlorine concentration at node 104, with four scenarios (1.5%) with values close to 2.0 mg/L and three nodes (1.12%) with approximate values at 3.0 mg/L. Finally, 10.78% of the nodes affect up to 1 mg/L.

Considering, for example, a leak at node 37, the chlorine concentration at node 104 is completely changed (Figure 8). Figure 8 shows that the chlorine concentration at node 104 assumes a completely different behavior depending on the presence of a leakage at node 37 or not. Although the maximum value changes slightly, passing from a 3.00 mg/L without leakage to a 3.05 mg/L with the leakage, the signal exhibits consistent variation during time. These variations may provide a significant source of information. For example, it can be observed that the concentration difference is 2.5 times greater than the sensitivity of the monitoring equipment presented by Gobet et al. [40] (0.02 mg/L). This would result in a possible detection by the opportune sensors.

Given that the purpose of this study is to support leak detection tasks, and assuming that a sensor with a sensitivity of 0.02 mg/L could be placed at node 104, the induced-changes in the chlorine concentration due to a leakage could be theoretically detected in 99.5% of network nodes. These monitoring data can be used coupled with data-driven techniques and hydraulic simulations so that, through them, information such as leak location, determination of leak demand, and chlorination failure may be rapidly and accurately identified and corrected.

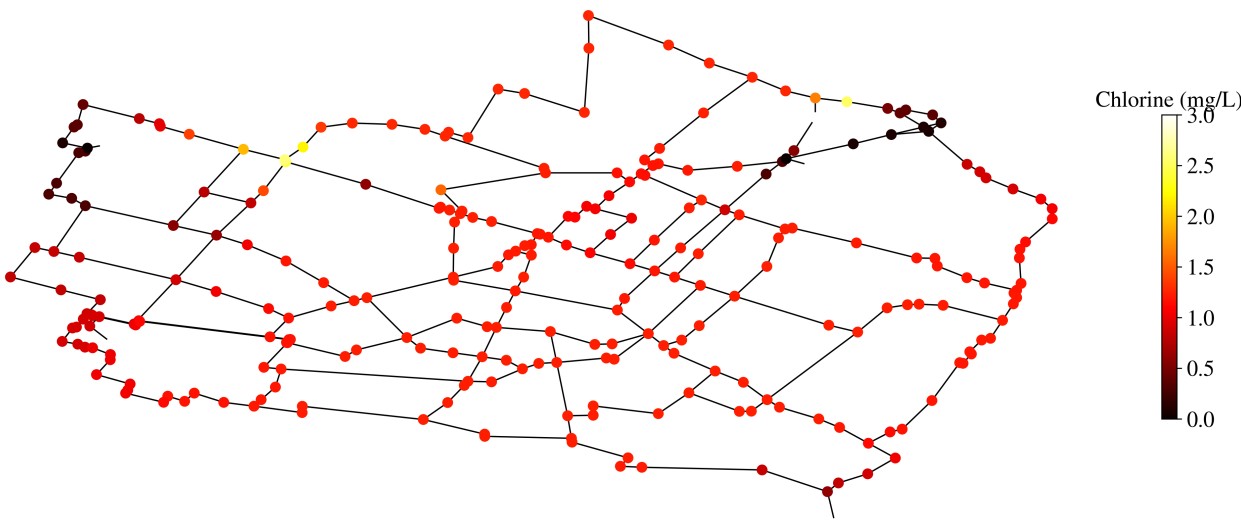

**Figure 7.** Maximum difference in chlorine concentration.

The important variability highlighted in Figures 7 and 8 was expected also for chlorine, because the mass transfer is altered when changing the pipe flow rates to meet the leak demands. To assess the changes inducedby leak events in the chlorine concentration values during the simulations, it is proposed to use the graph representation of the network. In particular, the following section provides an analysis of shortest path between the reservoir and node sensors.

4.2.2. Shortest Path and Flow Changes

For this application, the Modena network is represented as a graph where the nodes are represented by vertices, the pipes are represented by edges, and the maximum flow rates are used as edge weights. With these assumptions, Figure 9 shows the shortest path results, as well as the maximum flow rates differences for each pipe due to a leakage.

Figure 9a shows the shortest path found between reservoir 269 and node 104, with the entire total amount of water needed to supply the node's demand coming from reservoir 269. This total amount is determined through the Trace function of the Epanet software. Figure 9b shows the differences, at each pipe, between the maximum flow values without leakages and the maximum flow values with a leak in node 37. It can be noticed that there are changes of up to 2.5 L/s, and also that there are changes in the flow rates in the pipes present in the shortest path between the reservoir and node 104. Thus, these changes in the flow rates in the pipes that belong to the shortest path explain the variations highlighted in the quality parameters analyzed in Section 4.2. Another leak is, afterward, simulated individually at node 133, and Figure 10 shows the variations of chlorine concentration in node 104, as well as the map of the variations of the flow rates during this different leak.

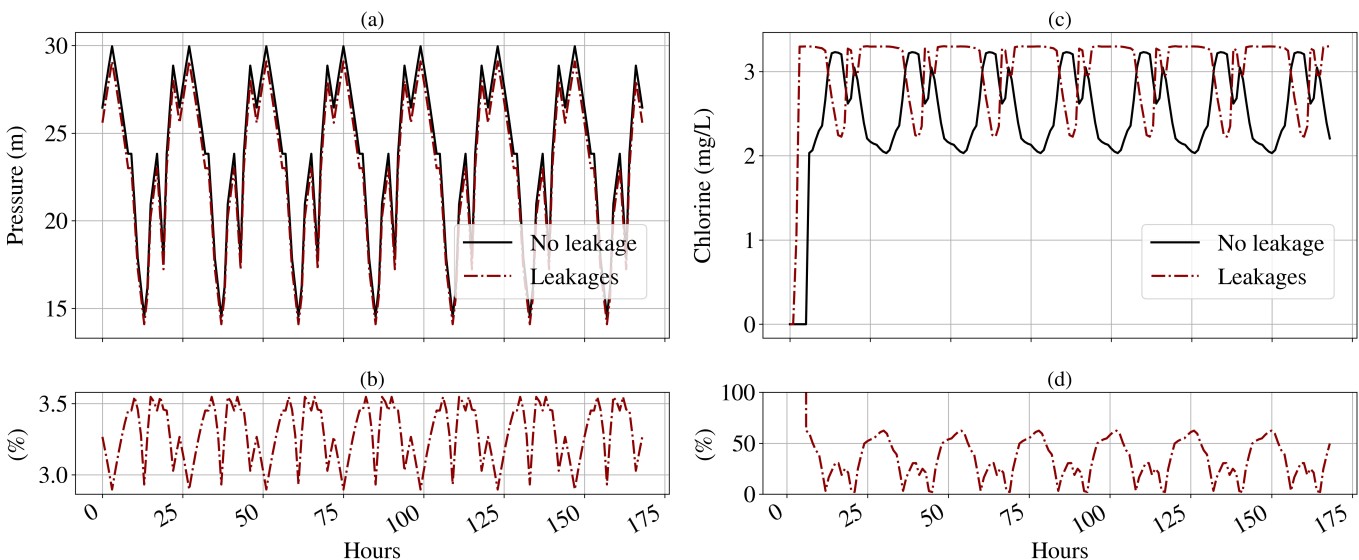

**Figure 8.** Pressure and chlorine behavior at note 104 with leakage at node 37: (**a**) Pressure variation, (**b**) Pressure variation delta, (**c**) Quality variation and (**d**) Quality variation delta.

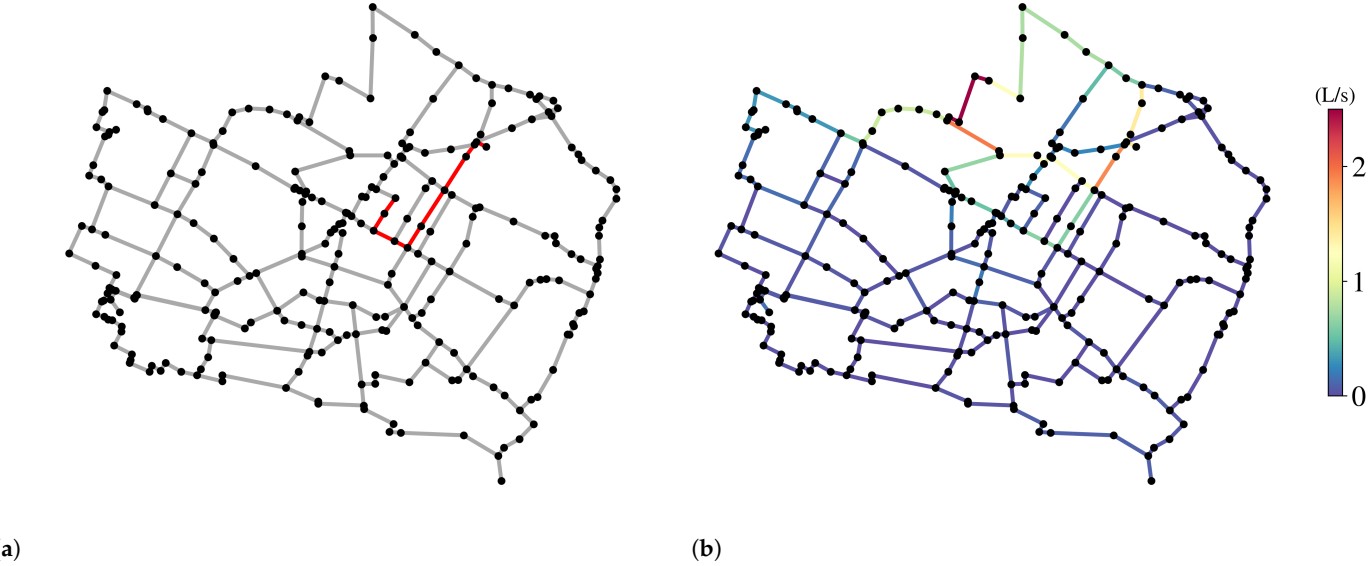

(**a**)

(**b**)

**Figure 9.** Network behavior flow: (**a**) Shortest path—reservoir 269 to node 104 and (**b**) flow differences—leakage node 37.

It can be observed in Figure 10 that, once again, the behavior of the chloride concentration at node 104 is altered, even with a leak simulation in a more distant node that receives a smaller contribution of water from reservoir 269. This behavior further supports the use of quality data for leak detection, since it was proven that points in the network can suffer great changes in quality, even with distant leaks, which perhaps would not affect the pressure, for example, of this node.

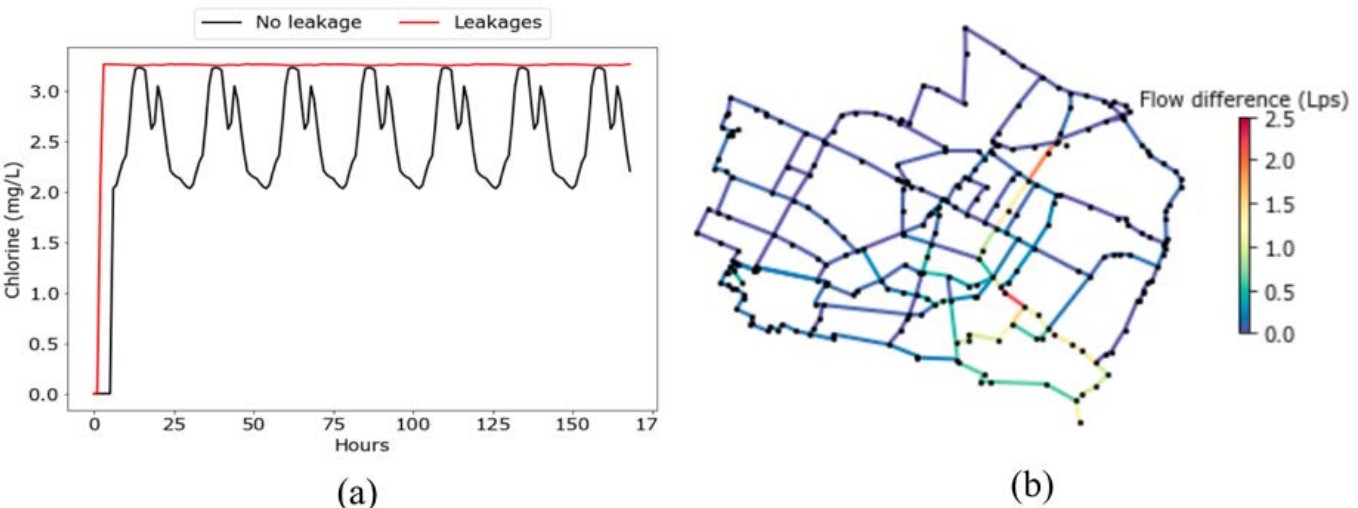

**Figure 10.** Behavior of quality and flow with leakage 133: (**a**) chlorine—node 104 and (**b**) flow differences—leakage node 133.

### 5. Conclusions

The water loss and contamination in the water distribution networks is a global problem that needs attention for the development of techniques that help to reduce such damages. Water quality data provide an abundant source of information for the leak location and optimization of the sensor placement in the water distribution network. In addition to the high sensitivity of such a parameter, the equipment for its monitoring is already on the market with a high accuracy, confirming the importance of exploring this aspect.

This research is also a proof of concept on water quality monitoring for leak situations. The results show, through computer simulations and the observation of water age and chlorine concentration, that the monitored parameters of water quality undergo changes greater than pressure (data usually used for detection) and prove to be an additional and effective source of information in the case of leaks. The change of water paths in the pipes to meet the leak demand modifies the entire behavior of the quality parameter changes, and in most cases, a significant change is observed. This change does not only occur mainly in nodes immediately connected to reservoirs or very close to them. Even so, it stands out in this proof of concept that water quality data can be used for various purposes, such as for detecting and locating leaks and failures in the operation of pumps and valves. The behavior of flows was proven by evaluating the shortest path, by representing WDS as a graph. Through a simulation of leaks in all nodes, it was possible to observe that water quality is strongly affected in some nodes. Thus, a small number of sensors placed in the WDN at optimized points, as happened with node 104, could mean a great efficiency in detection of leaks and great savings for service providers.

Determining how to monitor the chlorine concentration is effective for detecting leaks, and other factors may be explored, such as chlorination failures and pump and valve operation. An additional point that can also be explored is the quality sensors placement to detect and locate leaks.

**Author Contributions:** Conceptualization and methodology, B.B., D.B., A.Z. and I.A.; software and data analysis, D.B., A.Z. and I.A.; writing and preparation of the original draft, D.B., A.Z., I.A. and B.B.; visualization, B.B., G.M. and E.L.J.; supervision, B.B., G.M. and E.L.J. All authors have read and agreed to the published version of the manuscript.

**Funding:** This research was funded by National Council for Scientific and Technological Development 420 (CNPq) and Coordenação de Aperfeiçoamento de Pessoal de Nível Superior–Brasil (CAPES)–421 Finance Code 001.

**Institutional Review Board Statement:** Not applicable.

**Informed Consent Statement:** Not applicable.

**Data Availability Statement:** Not applicable.

**Acknowledgments:** The authors acknowledge the financial support from the National Council for Scientific and Technological Development (CNPq) and Coordenação de Aperfeiçoamento de Pessoal de Nível Superior – Brasil (CAPES) – Finance Code 001.

**Conflicts of Interest:** The authors declare no conflict of interest.

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
