# Peer review of "An Investigation on the Effect of Leakages on the Water Quality Parameters in Distribution Networks"

_water, doi:10.3390/w15020324_

Round 1

Reviewer 1 Report

General comments:

This article presents simulation results focusing on the effects on water quality caused by different leaks through computer simulations using EPANET software and the Water Network Tool for Resilience (WNTR) package in Python. This paper analyzes the water quality behavior during a spill event, in which a pipe network spill simulation is performed and water age and chlorine concentration are observed. In this paper reasonable and well-founded predictions are made based on the changes in water quality parameters associated with simulated leaks, thus making reasonable and well-founded predictions about leaks in municipal pipeline networks, which have some practical significance and application value. The experimental design is logical and appropriate, but there are still the following problems:

1)        Quality analysis is considered in this article as an important factor. Do changes in the external environment have an impact on this parameter? For example during peak water use periods.

2)        Since the location of leaks is random, how can we ensure that there is a detection device at every location where a leak occurs so that we can convey the leak information to the official website?

3)        When the leakage volume is small or the leakage volume of the pipe network is not enough to affect the change of water quality, how can the system make a judgment and prediction?

4)        Figure 1 is a map of the water supply network, suggesting that drinking water treatment plants and the end of the network can be marked, which will facilitate the reader to better understand.

5)        Figure 2 shows the sensitivity of pressure and quality at different locations on the official website, and it is suggested that the test nodes can be selected based on the distance from the test point to the user as the test node selection criteria, which can be used to further examine, the relationship between sensitivity and distance.

6)        Figure 7 is the maximum difference graph of chlorine concentration, we know that drinking water in the municipal network itself also has a tendency to reduce, especially with the time correlation is stronger, in this figure is there consideration of this factor?

7)        Chapter 4.2.2 tells about the shortest path and flow variation. Nowadays, the pipeline network is intertwined, and the nodes can be appropriately tilted to the key water transmission hubs when selecting the nodes.

Author Response

Dear reviewer,
thank you for your suggestions and corrections. Send attached the answers to your questions. 

Yours sincerely.

Daniel Bezerra Barros

Reviewer 2 Report

The paper presents the research about how water quality data can be used as a parameter for leak detection and location in the network using EPANET and WNTR.

A series of simulations are performed and the conclusions are the expected given the antecedents but I think that the more interesting and innovative issue can be that presented in section 4.2.2, that is the use of the shortest path and flow changes to detect the appearance of leaks. Unfortunately, equally it is shown that the change in the quality parameter indicates only the presence of leaks but not the location of them, which is evident from Figures 9 and 10: there is a change in flow in the shortest path even if the leak is in node 37 or node 133.

So, I do not know whether this investigation has the entity necessary to be published.

Morever, the title is somewhat confusing. The reader expects to find the answer to the question “May quality data help with leak detection?” when the answer is in the paper by Kumar et al (2010), reference [27].

Author Response

Dear reviewer,
thank you for your suggestions and corrections. Send attached the answers to your questions. 

Yours sincerely

Round 2

Reviewer 1 Report

The author needs to clarify the necessity and innovation of this study, as well as the reliability of the conclusions.
Morever, the title is somewhat confusing. The reader expects to find the answer to the question “May quality data help with leak detection?”

Author Response

Thank you again for your comments and suggestions. The main idea of the article is to present why, how and how much water quality parameters are changed in case of leaks. Based on that the hydraulic parameters are changed in the occurrence of leaks and that the quality parameters are strongly linked to hydraulics, when the changes in hydraulic parameters are also in the quality parameters. Therefore, the work presented here sought to understand and validate the behavior of water quality in case of leaks, once proven the changes in case of leaks, this new source of information beyond the data commonly used (pressure, flow and tank levels) can lead to better correlation between data (Ularu et al., 2012), assisting in various data analysis processes.

As innovation are presented as proof of concept these changes in water quality through leaks. This proposal was only aborted by Kumar et al. (2010), but the authors do not present case studies and results, only an initial idea. Another innovative proposal is to use the shortest path, considering the network and its connections as a graph, to analyze the changes in hydraulic parameters in the pipes that supply the leak demand, corroborating with the paper main idea.

The title of the article, abstract, introduction and conclusions of the manuscript are changed to better explain the main idea of this work.

Kumar, J., Sreepathi, S., Brill, E. D., Ranjithan, R., & Mahinthakumar, G. (2010). Detection of leaks in water distribution system using routine water quality measurements. In World Environmental and Water Resources Congress 2010: Challenges of Change (pp. 4185-4192).

Ularu, E. G., Puican, F. C., Apostu, A., & Velicanu, M. (2012). Perspectives on big data and big data analytics. Database Systems Journal3(4), 3-14.

Reviewer 2 Report

I still think that the more interesting and innovative issue can be that presented in section 4.2.2, that is the use of the shortest path and flow changes to detect the appearance of leaks

The title does no reflect the content of the study. A more appropriate title would be:

Detection of leaks in water distribution networks monitoring quality data using a method based on the flow shortest path.

Abstract

I think that the 3 first sentences should be removed and to be maybe placed in the Introduction.

Moreover, the abstract is missing the main conclusions of the study. Please, rewrite it.

 Figure 1: please include the position of the water treatment plant.

In conclusions

“Water quality monitoring proved to be more effective in detecting leakage than monitoring node pressures” I am not sure if this is a conclusion of this research or it is a known fact. Is this? If so, please make it more remarkable. If not, please include reference.

I recommend the authors to reinforce the main findings of this investigation, which is the method based in the shortest path, I think.

Author Response

Thanks again for the comments and suggestions. Section 4.2.2 deals with the observation of the shortest path between the reservoir and the leaking point, in terms of changing in the pipes flow rates. Anyway, it is an innovative analysis using graph theory on finding the shortest path, considering the network and its connections as a graph. However, the main idea of the article is not leak detection by this proposal, but to present and validate the changes that occur in water quality parameters in the presence of leaks, for better understanding how water quality data can be used on process of detecting and localizing leaks.  The proposal presents why, how and how much water quality is affected by leaks. Therefore, we used the proposal of presenting the shortest path to show the changes in the hydraulic parameters of the pipes that supply the leakage. Thus, as quality parameters are closely linked to hydraulics, when there is change in hydraulic parameters there is also in quality parameters.  Therefore, this paper is a proof of concept that presents a new source of information that can be used for various purposes, one of them being the leak detection.

A proposal that follows the same principle is presented by Kumar et al. (2010), in which the authors one exposes that water quality can be changed in case of leaks. However, the authors present only initial ideas, without presenting case studies and results. Furthermore, based on graph theory approach, it’s possible to explain clearly why water quality is changing at certain nodes. Therefore, this manuscript sought to understand and validate the behavior of water quality in case of leaks, once proven the changes in case of leaks, this new source of information in addition to the data commonly used (pressure, flow and tank levels) can lead to a better correlation between data (Ularu et al., 2012), assisting in various data analysis processes.

Finally, the Abstract and Conclusion are changed and corrections are accepted Regarding Figure 1, the authors who present the Modena network (Bragalli et al., 2012) do not inform the location of the water Treatment plant. Therefore, we use the reservoirs as a source of chlorination.

Kumar, J., Sreepathi, S., Brill, E. D., Ranjithan, R., & Mahinthakumar, G. (2010). Detection of leaks in water distribution system using routine water quality measurements. In World Environmental and Water Resources Congress 2010: Challenges of Change (pp. 4185-4192).

Ularu, E. G., Puican, F. C., Apostu, A., & Velicanu, M. (2012). Perspectives on big data and big data analytics. Database Systems Journal3(4), 3-14.

Bragalli, C.; D’Ambrosio, C.; Lee, J.; Lodi, A.; Toth, P. On the optimal design of water distribution networks: a practical MINLP approach. Optimization and Engineering 2012, 13, 219–246

Round 3

Reviewer 1 Report

Check the details of the article again

Reviewer 2 Report

Thank you for attending the suggestions.

Sincerely,